# Boosting Non-causal Semantic Elimination: An Unconventional Harnessing of LVM for Open-World Deepfake Interpretation

Submission Id: 1702

## ABSTRACT

The rapid advancement of generation methods has sparked significant concerns about potential misuse, emphasizing the urgency to detect new types of forgeries in open-world settings. Although pioneering works have explored the classification of open-world deepfakes (OW-DF), they neglect the influence of new forgery techniques, which struggle to handle a greater variety of manipulable objects and increasingly realistic artifacts. To align research with the evolving technologies of forgery, we propose a new task named Open-World Deepfake Interpretation (OW-DFI). This task involves the localization of imperceptible artifacts across diverse manipulated objects and deciphering forgery methods, especially new forgery techniques. To this end, we leverage non-casual semantics from large visual models (LVMs) and eliminate them from the nuanced manipulated artifacts. Our proposed model includes Semantic Intervention Learning (SIL) and Correlation-based Incremental Learning (CIL). SIL enhances the inconsistency of forgery artifacts with refined semantics from LVMs, while CIL combats catastrophic forgetting and semantic overfitting through an inter-forgery inheritance transpose and a targeted semantic intervention. Exploiting LVMs, our proposed method adopts an unconventional strategy that aligns with the semantic direction of LVMs, moving beyond just uncovering limited forgery-related features for deepfake detection. To assess the effectiveness of our approach in discovering new forgeries, we construct an Open-World Deepfake Interpretation (OW-DFI) benchmark and conduct experiments in an incremental form. Comprehensive experiments demonstrate our method's superiority on the OW-DFI benchmark, showcasing outstanding performance in localizing forgeries and decoding new forgery techniques. The source code and benchmark will be made publicly accessible on [website].

## KEYWORDS

Deepfake Detection, Interpretation, Open-world, No-casual Elimination, LVM

## 1 INTRODUCTION

As the realm of Artificial Intelligence Generated Content (AIGC) evolves, the creation and alteration of arbitrary objects become more feasible. Unfortunately, such technological advancements

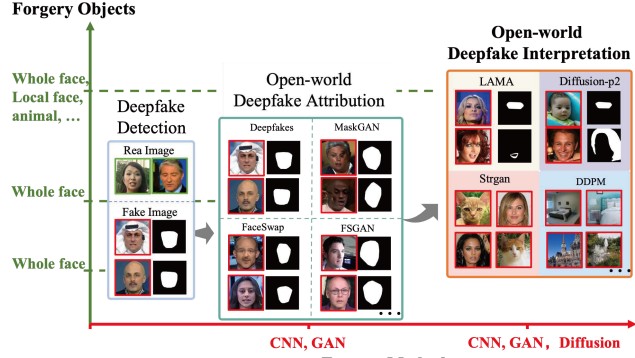

**Figure 1: The proposed OW-DFI task vs. existing deepfake detection tasks. OW-DFI covers a wider array of forgeries, crafted using a more sophisticated and varied range of forgery technologies, incorporating 24 AIGC forgery methods based on CNN, GAN, Diffusion, etc. Moreover, acknowledging the advancements in generative capabilities, OW-DFI offers a more comprehensive assortment of forgery subjects compared to its predecessor, OW-DFA and DFD. This enhancement is not limited to the entire facial area but also includes specific face-related objects like eyeglasses, and extends to non-facial elements such as animals and backgrounds.**

also provide malicious entities with tools to manipulate images for nefarious purposes, such as swaying public opinion or fabricating evidence. Thus, the development of robust forgery detection technologies becomes essential to provide expert testimony and protect information security, public sentiment, and societal trust.

Current methods in forgery detection [14, 23] show promise in identifying falsified images with considerable accuracy. However, they often fall short in offering nuanced interpretations of subtle forgeries. This gap has led to the increased focus on Deepfake Attribution (DFA) [38] and Deepfake Localization (DFL) [31, 51], methodologies acclaimed for their ability to detail the "*how*" and "*where*" behind digital manipulations. The prevalent approaches for deepfake interpretation generally presuppose a static environment, assuming identical forgery distributions across training and testing datasets. With AIGC technology's continuous evolution, adapting detection mechanisms to uncover novel forgery techniques in real-world scenarios becomes crucial.

Until now, the exploration into open-world deepfake (OW-DF) detection has been relatively limited. A significant development is the introduction of open-world deepfake attribution (OW-DFA) [38], which aims to identify undisclosed forgery methods used on unlabeled facial imagery. While this pioneering work establishes a basis for comprehending OW-DF, its emphasis on classification tends to

marginalize the intricate visual manipulations brought about by emerging forgery techniques. Our study delves into two pivotal dimensions of these evolving visual effects: **an extensive range of manipulable objects** and **increasingly nuanced forgery signatures**. This expansion in manipulable objects is made possible by the advanced capabilities of AIGCs to inpaint any subject with specific content or modify backgrounds with ease [33, 40]. However, OW-DFA [38] primarily focuses on facial characteristics, such as identity and expressions, hence overlooking an extensive spectrum of manipulable elements, ranging from particular body parts to accessories and animals. Pertaining to the challenge of detecting nearly invisible forgery traces, it introduces a critical inquiry: "*Where exactly is the forgery located?* Given these considerations, the setting of OW-DFA is deemed insufficient in interpreting forgeries amidst the continually progressing visual effects in an open-world context. To bridge this gap, we propose the novel Open-World Deepfake Interpretation (OW-DFI) task, aimed at enhancing OW-DF by providing a comprehensive interpretation of a wide array of realistically manipulable objects, as illustrated in Figure 1. This interpretation process entails the precise localization of all forgery regions and trustworthy attributing both known and newly discovered forgery methods. Furthermore, considering the substantial visual gap between old and advanced forgeries, our OW-DFI task is devised to operate in an incremental manner, addressing the continual evolution of AIGCs within the open-world paradigm.

OW-DFI typically focuses on learning representative semantic features to identify out-of-distribution (OOD) data and classify objects that are introduced incrementally. OW-DFI faces unique challenges in identifying forgery traces, as the embeddings of subtle artifacts involved in forgeries are susceptible to being overshadowed by more dominant semantic features. The distinction highlights the specialized nature of OW-DFI in dealing with the intricacies of forgery detection. This limitation arises from the inadequacy of training samples to comprehensively represent the diversity of manipulable objects, potentially resulting in overfitting to common semantic features such as "hair". To tackle this challenge, we propose a Deepfake Interpretation Network (DIN), tailored to eliminate non-causal semantics within OW-DFI. Specifically, we propose a semantic intervention learning that harnesses the rich semantics from LVMs. By adaptively refining semantic-invariant channels within the overall feature space derived from an arbitrary LVM, our proposed semantic purification module effectively removes ambiguous information without introducing additional parameters. Furthermore, we introduce a semantic-prior intervention module to inhibit the model's acquisition of non-causal semantics. To fortify lifelong learning capacities, we integrate correlation-based incremental learning to consider the inheritance between old forgery technologies and advanced ones. Additionally, to mitigate overfitting to semantics with few samples, we propose a semantic-prior orientation module to dispel coherence with semantics within the confines of each unique novel class. In summary, our contributions are three-fold:

- We present a novel task called Open-World Deepfake Interpretation (OW-DFI), tailored to tackle the challenges of interpreting and identifying new forgeries in an incremental framework. This task is devised to counteract the sophisticated visual effects and the expanded variety of manipulable objects that arise with the advent of new forgeries in open-world environments.

- We propose a Deepfake Interpretation Network (DIN) to localize deepfakes and investigate the specific forgery method. In scenarios involving previously unseen forgeries, DIN distinguishes it from the known ones. The proposed network harnesses the intricate semantic representations from large visual models and consists of a semantic purification module as well as a semantic intervention module to suppress the learning of non-causal semantics. In its incremental learning phase, DIN adopts a correlation-based incremental module, facilitating knowledge transfer across various forgeries, and incorporates a semantic-prior orientation module to counteract semantic overfitting.

- Through comprehensive experiments conducted on 24 forgery methods collected from 4 diverse datasets, including ForgeryNet, HiFi-IFDL, Dolos, and FF++, we demonstrate the outperforming performance of our approach to localize deepfakes and discover novel forgeries, which provides reliable interpretation.

## 2 RELATED WORK

### 2.1 Forgery Interpretation

To combat the abuse of AIGC, there is an escalating demand for forgery detection methods that can effectively identify manipulated images. With the advancement of AIGCs, several approaches[30, 43, 45] have been developed to extract universal forgery features across different forgery methods. For example, DFIL [30] refines forgery features from emerging samples to execute generalized binary classification within a continuous learning paradigm. As AIGC-generated artifacts grow increasingly indistinguishable from genuine content, there is a heightened demand for more nuanced interpretation. Existing forgery interpretation methods can be broadly categorized into two categories: forgery attribution, which seeks to ascertain the source model of counterfeit images, and forgery localization, which aims to identify the manipulated region.

***Forgery attribution.*** Previous studies [46] have largely concentrated on attribution within fully synthesized GAN models by applying intricate tactics to identify unique 'fingerprints' left by various network architectures. For instance, DNA-Det [46] enhances the discrepancy between various forgery techniques through detailed patch-level contrastive learning. Facing the rapid progression of GAN technologies, Open-world GAN [12] undertakes the categorization of both recognized and newly emerging GAN methods for attribution purposes. Additionally, CPL [38] extends fully synthesized images with facial region manipulation, facilitating actions like identity swapping and expression transferring. Nonetheless, the increasing diversity of manipulable objects accompanying the evolution of AIGCs has not garnered adequate attention. Therefore, solely focusing on the question of "*What constitutes image forgery*" is insufficient.

***Forgery Localization.*** Initial efforts [13, 39] in localizing manipulated pixels have targeted detecting inconsistencies in texture. For example, TurFor [13] merges the original RGB modality with a noise-sensitive fingerprint to amplify the inconsistency of manipulated regions. SAFL [39] aims to distinguish patches sharing identical semantics within an image to highlight semantic-independent inconsistencies. However, these methods are mainly effective against

manual editing like copy-move and splicing, potentially struggling against advanced generation techniques that yield uniform textures. To pinpoint CNN-based forgeries, LVNet[35] proposes a multi-stage modality fusion model that incorporates multi-scale information for forgery localization. DADF [19] employs a multi-scale adapter to capture both short-range and long-range forgery contexts from pre-trained LVMs. However, while these localization methods prove effective in controlled environments, their adaptability to novel forgery techniques and the broadening spectrum of manipulable objects in an open-world setting remains uncertain.

## 2.2 Open-World Settings

The concept of *Open-world* [2] refers to a model that performs trust-worthy classification and incrementally learns to identify newly introduced classes. Benefiting from its resemblance to real-world scenarios, research on open-world problems has significantly increased [3, 10, 25]. In the forgery domain, the recent emergence of OW-DFA [38] marks the first framework with open-world forgery attribution. However, OW-DFA focuses solely on exploring new forgery categories within the open classification space, neglecting the interpretation of open visual effects, which encompass a wider array of realistic and diverse manipulable objects. Considering this, we propose OW-DFI with the aim of providing pixel-level trustworthy forgery attributions and achieving novel forgery discovery under the challenging few-shot incremental setting. Unlike the learning of salient semantics in other visual tasks, the subtle nature of forgery traces introduces two key challenges that increase the complexity of OW-DFI. During the feature learning phase, forgery features are prone to being overshadowed by more prominent semantic features. Some methods [14, 23, 39] attempt to isolate semantic features by training additional semantic models for fixed categories, presuming uniform semantics across training and testing objects. Nonetheless, the occurrence of arbitrarily forged objects in the open world poses significant challenges. To address these issues, we propose exploiting the extensive semantic information encapsulated in LVMs such as CLIP [32] and DINO [29]. During the incremental learning phase, a prevalent strategy [3, 10] involves employing prototype-based metric learning to mitigate catastrophic forgetting. However, the subtle discrepancies between forgery methods pose challenges in explicitly retaining the forgery knowledge of the model. In the field of OW-DF, limited attention has been paid to incremental few-shot learning [30, 47], which lacks interpretability as it primarily updates the binary classification space, neglecting the nuanced relationships between forgery methods. In OW-DFI, we investigate the knowledge transmission from established forgeries to more advanced ones, thereby developing an attribute space to improve interpretability.

## 3 OPEN-WORLD DEEPFAKE INTERPRETATION TASK

The OW-DFI task aims to address challenges posed by both an open classification space and the increased diversity of realistic manipulable objects resulting from the evaluation of AIGCs. Its pipeline involves the comprehensive interpretation of known forgeries and the discovery of novel ones, supplemented by incremental few-shot learning for continuous assessment. Initially, the pipeline

| Categories | Method | Source | Objects | Num. samples | | |
|---|---|---|---|---|---|---|
| | | | | Train | Val | Test |
| CNN | Faceshifter[21] | ♠♣ | 1 | 1500 | 300 | 600 |
| | Deepfakes | ♡♣ | 1 | 2800 | 554 | 880 |
| | Face2Face[42] | ♡ | 1 | 2950 | 586 | 568 |
| | NeuralTextures[41] | ♡ | 1 | 5 | 0 | 578 |
| | FaceSwap | ♡ | 1 | 5 | 0 | 582 |
| GAN | FirstOrderMotion[36] | ♠ | 1 | 1485 | 300 | 300 |
| | STGAN[24] | ♣ | 6 | 1017 | 900 | 900 |
| | MaskGAN[20] | ♠ | 6 | 1482 | 300 | 900 |
| | SC-FEGAN[17] | ♣ | 1 | 5 | 0 | 300 |
| | FSGAN[28] | ♣ | 1 | 5 | 0 | 300 |
| | ATVG-Net[4] | ♠ | 1 | 5 | 0 | 300 |
| | Talking-headVideo[11] | ♠ | 1 | 5 | 0 | 300 |
| | StyleGAN2[18] | ♠ | 1 | 5 | 0 | 300 |
| | STarGAN[7] | ♣♠ | 2 | 5 | 0 | 600 |
| | DiscoFaceGAN[9] | ♣ | 1 | 5 | 0 | 300 |
| | DSS[15] | ♣ | 1 | 5 | 0 | 300 |
| | HiSD[22] | ♠ | 1 | 5 | 0 | 300 |
| Diffusion | repaint-p2[6] | ♢ | 5 | 3000 | 600 | 1200 |
| | pluralistic[48] | ♢ | 5 | 5 | 0 | 600 |
| | DDPM[16] | ♣ | 3 | 5 | 0 | 300 |
| | DDIM[37] | ♣ | 3 | 5 | 0 | 300 |
| | GLide[27] | ♣ | 1 | 5 | 0 | 300 |
| | D-latent[33] | ♣ | 3 | 5 | 0 | 300 |
| | ldm[33] | ♢ | 5 | 5 | 0 | 600 |

Table 1: Specification of the OW-DFI benchmark. It outlines 24 forgery methods and describes their types, originating datasets, categories of manipulated objects, and the number of samples. The source datasets are denoted as ♡ :=FF++, ♣ :=HiFi-IFDL, ♠ :=Forgerynet, and ♢ :=Dolos. These methods are organized into base sets and incrementally joint sets across  Session 1 ,  Session 2 ,  Session 3 , and  Session 4 , reflecting their chronological order of public availability.

utilizes a base dataset $D^{in} = D_0^{in}, D_1^{in}, D_2^{in}, ..., D_N^{in}$, comprising a set of real images $D_0^{in}$ and $N$ base forgery methods with sufficient annotations. Additionally, $D^{out} = D_1^{out}, D_2^{out}, ..., D_M^{out}$ represents $M$ subsequent forgery methods collected over time as new forgery technologies emerge. Following training on $D^{in}$ as the initial deepfake interpretation task, the model undergoes incremental updates through sequential training with few-shot annotations from $D^{out}$. Let $i$ denote the $i^{th}$ task with known forgeries $D^i = \{D_0^{in}, D_1^{in}, D_2^{in}, ..., D_N^{in}, D_1^{out}, ..., D_i^{out}\}$. The model incrementally learns from $D^i$ to $D^{i+1}$ as new forgeries $D_{i+1}^{out}$ emerge, aiming to interpret updated known methods and discover more advanced ones.

To simulate the diverse manipulation styles exhibited by various forgery methods, we establish the OW-DFI benchmark based on 4 widely utilized forgery localization datasets, including ForgeryNet [15], HiFi-IFDL [31], Dolos [51], FF++ [34]. These datasets offer extensive data coverage and encompass a wide range of manipulation methods. As shown in Table 1, the OW-DFI benchmark encompasses 24 forgery technologies spanning 6 types of forgery objects, including the entire face, nose, mouth, eyebrows, hair, and even animals like cats. These technologies are categorized into three forgery categories, each further subdivided into base methods, which have sufficient samples, and novel methods (highlighted in colors), which have limited samples and are designated for incremental learning. To replicate the evolutionary progression of forgery technologies, we partition the 17 novel methods into 4 sessions based on their

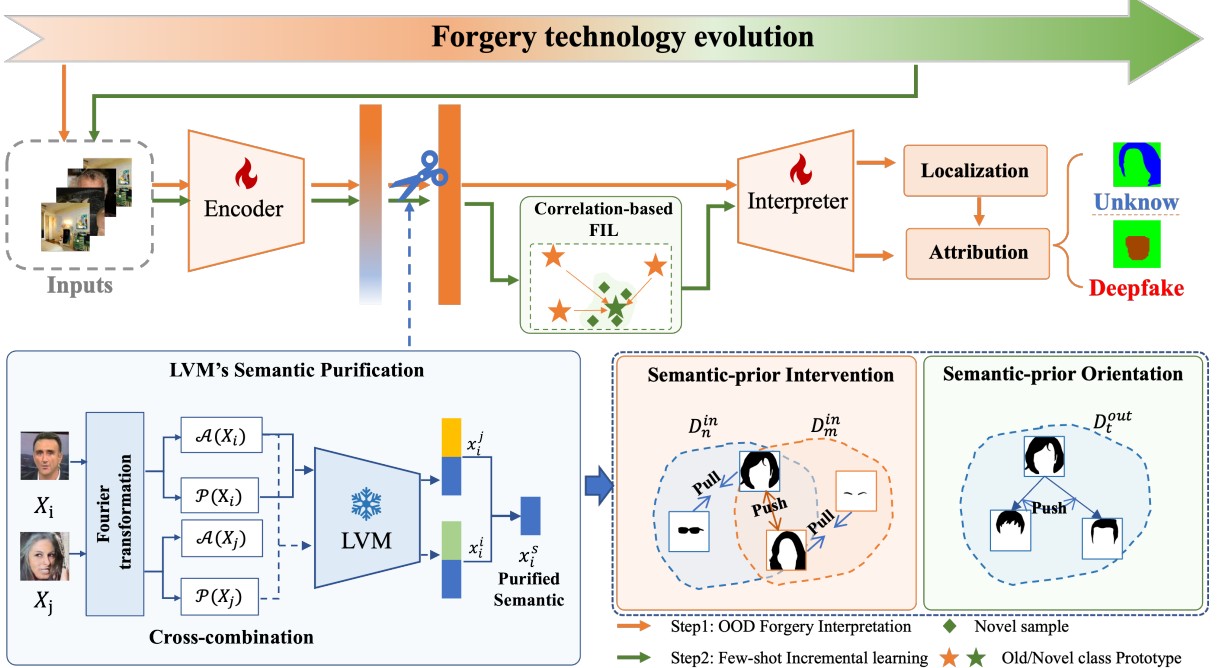

**Figure 2: The architecture of our Deepfake Interpretation Network (DIN). It is designed to adapt to and evolve with advancements in forgery technologies, encompassing two main phases: 1) Step 1 (depicted by orangeish arrows) is dedicated to conducting known forgery interpretation and unveiling novel forgeries. 2) Step 2 (depicted by greenish arrows) addresses the adaptation to forgery technologies through the proposed correlation-based few-shot incremental learning. Additionally, we purify semantics from LVMs and utilize them to eliminate non-causal semantics through two distinct modules: the semantic-prior intervention module for Step 1, and the semantic-prior orientation module for Step 2.**

public release times, each represented by different colors. Additionally, we integrate a randomly sampled pool of 11,499 real faces sourced from the authentic sets of these four datasets.

## 4 DEEPFAKE INTERPRETATION NETWORK

### 4.1 The Architecture

Figure 2 illustrates the architecture of our proposed Deepfake Interpretation Network (DIN), comprising two primary components dedicated to deepfake interpretation: a forgery feature encoder $\Theta$ and an interpreter $f$. To address the requirements for generality in localization and discrimination in attribution, we partition the classifier into two sequential branches, including the localization branch $f_l$ and the attribution branch $f_a$. Formally, given an input image $X \in \mathbb{R}^{3 \times H \times W}$, the encoder $\Theta$ extracts forgery features $x^e \in \mathbb{R}^{C \times H \times W}$ through stacked convolution layers. Then $f_l$ processes $x^e$ to produce localization results $Y_l \in \mathbb{R}^{H \times W} = f_l(x^e)$, which classify each pixel as real or forged. For attribution, traditional learnable classifiers are less effective as they tend to allocate all feature space to known forgeries and may fail to detect novel types. Instead, we employ a metric-based approach [3, 10] to determine forgery type based on similarity to open method prototypes $f_a$. We start by incorporating a dimension reduction layer to derive class-specific features $x^a \in \mathbb{R}^{C' \times H \times W}$ from $x^e$. Considering the semantic locality of forgery, we further divide the localized

forgery into distinct local regions using Sobel edge division [10]. DIN then produces attributions for each local region by assessing the cosine similarity between the average-pooled region features $x^r \in \mathbb{R}^{K \times C'}$ and the momentum-updated forgery prototypes $P_t = \{p_k \in \mathbb{R}^{1 \times C'} | k \in D^t\}$.

To optimize, we minimize the localization loss $\mathcal{L}_{loc}(Y_l, \hat{Y})$, which guides the generation of pixel-level forgery localization, and the classification loss $\mathcal{L}_{cls}(Y_a, \hat{Y})$ for subsequent attribution. Additionally, to address the challenge of emerging novel forgeries across a broader array of objects, we focus on minimizing semantic interference in an incremental setting. To achieve this, we conduct LVM-based semantic intervention learning in the first interpretation step (depicted by orangeish arrows), elaborated in Section 4.2, and correlation-based incremental learning in few-shot incremental steps (depicted by greenish arrows), detailed in Section 4.3.

### 4.2 Semantic Intervention Learning

With the development of AIGC, depictions of complex visual effects now encompass both realistic and arbitrarily faked objects, which may hinder the learning of subtle forgery features embedded in salient semantics. To address this issue, we implement Semantic Intervention Learning (SIL) to disentangle and eliminate non-causal semantics by leveraging semantics derived from LVMs. Compared with previous methods [30] with semantic networks specifically

tailored to the training set, SIL not only obviates the necessity for additional parameters but also enhances the efficiency in learning the semantics of arbitrary objects.

**Semantic Purification Module.** Considering that LVMs are developed using diverse strategies and poses an inherently *black-box* nature, their feature space encompasses both semantic and ambiguous elements. Consequently, we propose a semantic purification module (SPM) to facilitate the process of intervening in forgery learning. Unlike prior approaches [19] that sought to unearth forgery features within LVMs, our SPM adopts an unconventional strategy by aligning with the common semantic training objectives across various LVMs.

Given a set of $N$ images, the extracted LVM features are denoted as $x^s \in \mathbb{R}^{N \times D}$, where $D$ represents the total number of channels. Our objective is to employ a purifier $P \in \mathbb{R}^D$ to isolate channels that are particularly relevant to semantics through the operation $\hat{x}^s = P \cdot x^s$. This selection process focuses on incorporating channels into $P$ that are: 1) invariant to semantic-preserving disruptions, thus discarding ambiguous information, and 2) rich in semantic content.

To fulfill the first criterion, we generate semantic-preserving samples that vary only in their forgery characteristics. Drawing on the Fourier transformation (FT) decomposition attributes [26, 44], where the phase component $\mathcal{P}(X)$ retains high-level semantics and the amplitude component $\mathcal{A}(X)$ encompasses low-level statistics such as forgery details, we modify amplitude information while maintaining phase information, as shown in Figure 2. Specifically, we perform linear interpolation between the amplitude components of different samples $X_n$ and $X_m$, which are manipulated by distinct forgery methods, as represented as Equation 1.

$$\hat{\mathcal{A}}(X_n) = (1 - \lambda)\mathcal{A}(X_n) + \lambda\mathcal{A}(X_m), \tag{1}$$

where $\lambda \sim U(0, 1)$. Following this, we recombine the modified amplitude component with the original phase component to produce a disrupted image $X_n^m$ via the inverse FT as shown in Equation 2.

$$X_n^m = \hat{A}(X_n) \times e^{-j \times \mathcal{P}(X_n)}. \tag{2}$$

To further refine the semantic-relative channels, we aim to remove those that demonstrate variability across the disrupted samples. For the $k^{\text{th}}$ channel, its semantic stability is determined using the variance criterion $V_k$ as defined as Equation 3.

$$V_k = \frac{1}{M} \sum_{m=0}^{M} (x^{ms}_k - \bar{x}^s_k)^2, \quad \bar{x}^s_k = \frac{1}{M} \sum_{m=0}^{M} x^{ms}_k. \tag{3}$$

In addition, the sufficiency of semantic information for the $k^{\text{th}}$ channel is assessed through its L1-normalization score, serving as the sufficiency criterion $S_k$. Furthermore, to evaluate the semantic richness of the $k^{\text{th}}$ channel, we assess its L1-normalization score, serving as the semantic sufficiency criterion $S_k$. After calculating these metrics, we combine $V$ and $S$ to finalize the channel's value, expressed as $P_k = V_k - S_k$. We then select the top-$Q$ channels with the smallest $P_k$ values as the purified semantics, denoted as $\hat{x}^s$. This procedure effectively purifies the feature space by eliminating superfluous and ambiguous channels, thereby enhancing the extraction of relevant semantic features from the LVM-encoded data.

**Semantic-prior Intervention Module.** Upon deriving the purified semantics, our objective shifts to curbing the non-causal semantic features within the forgery detection space. Inspired by contrastive learning [39], we propose to reduce the affinity of learned forgery features to purified semantics.

Intuitively, we begin by computing the forgery feature $f^a$ from $\hat{x}^a$ using masked average pooling guided by forgery mask $\hat{Y}$. Our methodology seeks to bring images manipulated by identical forgery methods closer together, thus prioritizing forgery-specific features over semantic content. This involves compiling a set of samples into the positive set $P(X_a) = \{X_i | \hat{Y}_i = \hat{Y}_a\}$, which share the same forgery label as the anchor image $X_a$. Concurrently, our strategy includes pushing samples altered by divergent methods away, particularly those exhibiting similar semantics. For this purpose, we identify a corresponding negative sample $\tilde{X}_i$ for each $X_i$ within $P(X_a)$ with the highest semantic similarity to $X_i$. The essence of our approach is to adjust the cosine similarity between the anchor sample $X_a$ and each positive sample $X_i$, ensuring it surpasses that between the anchor and its corresponding negative sample $\tilde{X}i$ by a margin $m$, as calculated in Equation 4.

$$\mathcal{L}_{sil} = \frac{1}{|P(X_a)|} \sum_{X_i \in P(X_a)} max(0, cos(f_a^a, \tilde{f}_i^a) - cos(f_a^a, f_i^a) + m). \tag{4}$$

By minimizing $\mathcal{L}_{sil}$, we enhance the development of a forgery feature space that is not only discriminative but also semantically distinct, thereby improving the precision of forgery attribution.

## 4.3 Open-World Deepfake Interpretation

In the open world, enhancing the capabilities of few-shot incremental learning is essential for effectively identifying evolving forgeries and uncovering new ones. Given the subtle nature of forgeries and the variability of objects that can be manipulated, few-shot incremental learning faces significant challenges in acquiring novel forgery embeddings. This process is often compromised by the interference of prominent semantic features, as few-shot training samples may not fully represent the diverse range of manipulated objects found in real-world scenarios. To address these challenges, we proceed with correlation-based incremental learning that leverages the inherited relationships among forgery techniques to mitigate semantic overfitting. Furthermore, to minimize semantic disturbances in the learning process, we introduce a Semantic-prior Orientation Module (SOM) by locally diminishing the alignment of forgery attributes with semantics.

**Correlation-based Incremental Module.** During the development of AIGCs, researchers often craft novel techniques inspired by existing methodologies. Therefore, we propose to exploit the correlation between AIGCs to initialize the learning of novel forgeries. Inspired by the optimal transport (OT) mechanism in information transmission [50], we employ the relationship between traditional and emerging classes as a cost function $C$, facilitating the transference of prior knowledge to new prototypes. In the $i^{\text{th}}$ session, the training dataset $D^{out}i$ comprises few-shot samples manipulated by innovative forgery methods. The cost function $C$ calculates the pairwise Euclidean distance between the previously established forgery prototypes $P_{i-1}$ and the features of novel classes in $D^{out}i$. Formally,

the correlation score from the $n$-th old prototype $p_n \in P_{i-1}$ to the $i$-th novel forgeries is calculated as in Equation 5.

$$C_{n,i} = (p_n - \frac{\sum_{k=1}^{|D_i|} \mathbb{I}(Y_k = i) x_k^a}{\sum_{k=1}^{|D_i|} \mathbb{I}(Y_k = i)})^2, \tag{5}$$

where a higher score implies a lesser transmission of old information to the novel forgery. By using older prototypes to initialize the prototype of the novel forgery method, we aim to counteract the semantic overfitting typically resulting from directly averaging novel features, which may otherwise exhibit closely related manipulated semantics.

*Semantic-prior Orientation Module.* During incremental learning, the intertwining of non-causal semantics with learned forgery features presents a substantial challenge, further exacerbated by the scarcity of training samples. Traditional methods of disentanglement in the global feature space might compromise the distinctiveness of existing forgery prototypes. Drawing inspiration from orientation distribution learning, we propose Semantic-Prior Orientation Loss (SOL), which fine-tunes the distribution of forgery features locally to diverge from the purified semantic features of LVMs.

As shown in the left corner of Figure 2, the procedure begins by selecting a sample pair $X_a$ and $X_r$ from the new dataset $D^{out}_i$, with $X_a$ serving as the rotation anchor and $X_r$ as the reference. For each reference $X_r$, we identify an additional sample $X'_r$ sharing the closest semantic resemblance. We then calculate the forgery vectors relative to the anchor for both the reference and its semantically similar counterpart as $\vec{v}^a_{ar} = x_a^a - x_r^a$ and $\vec{v}^a_{ar'} = x_a^a - x_{r'}^a$, respectively. The goal is to orient the vector $\vec{v}^a_{ar'}$ towards a semantically-weighted reverse distribution compared to $\vec{v}^a_{ar}$, achieved by minimizing Equation 6.

$$\mathcal{L}_{fsi} = \frac{1}{M} \sum_{i=0}^{|D^{out}_i|} \sum_{r \neq i, p} cos(x_p^s, x_r^s)(1 + cos(\vec{v}^a_{ar}, \vec{v}^a_{ar'})). \tag{6}$$

This strategy ensures that feature embeddings predominantly align in directions that contradict semantics, facilitating the disentanglement of non-causal semantics. Moreover, SOL specifically targets samples affected by identical forgery methods, enabling semantic removal within distinct novel class spaces without disrupting pre-existing knowledge.

# 5 EXPERIMENTS

## 5.1 Implementation Details

We implement the proposed approach using PyTorch. Our feature extractor is based on DeeplabV3+ [5], following the setup outlined in [10]. This setup includes the addition of two convolutional layers designated as $f_l$ for pinpointing the forgery region and another convolutional layer, $f_a$, to extract forgery attributes for the purpose of attribution. In the preprocessing stage, for images containing faces, we utilize Retinaface [8] to extract the face area, which we resize to 1.3 times the size of the face. Subsequently, we resize all input images to 224×224 pixels. Our experiments are conducted on a single RTX 3090 GPU and adopt SGD optimization with a batch size of 64. The initial learning rate is set at 0.1 and is methodically reduced following a power schedule of 0.9 with the PolyLR strategy.

| Method | Unknown | | | Known |
|---|---|---|---|---|
| | AUROC↑ | AUPR↑ | FPR↓ | mACC↑ |
| LVNet (MM′23)[35] | 19.62 | 29.95 | 97.59 | 87.32 |
| POP (CVPR′23)[25] | 22.56 | 29.78 | 97.30 | 88.05 |
| OW-DFA (ICCV′23)[38] | 75.64 | 68.05 | 70.58 | 86.42 |
| **CLIP-DIN (ours)** | 85.86 | 72.10 | 37.39 | 89.37 |
| **DINOv2-DIN (ours)** | **87.96** | **75.76** | **35.53** | **89.42** |

**Table 2: The performance comparison for forgery interpretation. All models are trained on the base set and evaluated on forgeries manipulated by both 7 seen and 17 unseen methods. CLIP-DIN and DINO-DIN represent the utilization of pre-trained models CLIP and DINOv2, respectively. The best result is marked in bold, and the second-best result is underlined. This notation is maintained across subsequent tables.**

For generating semantic features, we engage CLIP [32] equipped with a ResNet50 architecture and Dinov2 [29] using the ViT-L architecture. The foundational loss functions comprise the FocalTversky Loss [1] as the localization loss ($\mathcal{L}_{loc}$) and the cross-entropy loss for classification ($\mathcal{L}_{cls}$). The margin $m$ used in Equation 4 is set to 0.4. The loss function tailored for the first step is computed as $L_{base} = \mathcal{L}_{loc} + \mathcal{L}_{cls} + \lambda_{sil} * \mathcal{L}_{sil}$. To accommodate 5-shot incremental learning in successive sessions, the loss formula is modified to $L_{base} = \mathcal{L}_{loc} + \mathcal{L}_{cls} + \mathcal{L}_{fsi}$.

## 5.2 Benchmark Evaluation

The evaluation of the proposed DIN encompasses two key aspects, including the open-world deepfake interpretation and the few-shot incremental setting.

*Results of open-world deepfake interpretation.* We first evaluate our proposed DIN for open-world forgery localization and attribution. All methods are trained on $D^{in}$, which includes 7 forgery methods, and are tested across all 24 forgery methods. As for evaluation metrics, AUROC, AUPR, and FPR scores assess the model's ability to distinguish unknown forgeries by comparing the attribution results of known and unknown forgeries, whereas ACC provides a measure of pixel-level attribution accuracy for known forgery methods. The experimental results are reported in Table 2, where it is compared with several state-of-the-art (SOTA) methods.

The first row presents the interpretation results of LVNet [35] using the Maximum Softmax Probability (MSP) to identify unknown forgeries. It demonstrates robust performance on known forgeries (mACC: 87.32%) but shows a significant drop in accuracy for unseen forgeries. This suggests that conventional deepfake interpretation methods struggle to differentiate unseen forgeries from authentic or previously known forgery techniques. We also conduct comparisons with other open-world oriented methods such as POP [25] and OW-DFA. POP, which relies on specialized binary classifiers for different forgery techniques, exhibits inferior performance in identifying unseen forgeries, thus highlighting the importance of understanding interconnections between different forgery techniques. OW-DFA achieves an AUROC of 75.64, whereas our DIN attains 87.96. DIN excels in detecting regions of novel forgery, achieving a 12.32% improvement in AUROC over OW-DFA, thereby emphasizing its strength in distinguishing between unknown and known forgeries. Additionally, our DIN also obtains a 35.05% reduction in

| Method | Session1 | | | | Session2 | | | | Session3 | | | |
|---|---|---|---|---|---|---|---|---|---|---|---|---|
| | Known | | Unknown | | Known | | Unknown | | Known | | Unknown | |
| | $IOU_{novel}$ | $IOU_{old}$ | $IOU_{real}$ | FPR $\downarrow$ | $IOU_{novel}$ | $IOU_{old}$ | $IOU_{real}$ | FPR $\downarrow$ | $IOU_{novel}$ | $IOU_{old}$ | $IOU_{real}$ | FPR $\downarrow$ |
| LVNet (MM'23)[35] | 0.60 | 17.74 | 58.97 | 97.82 | 1.02 | 7.78 | 56.72 | 84.11 | 5.61 | 5.86 | 55.10 | 62.52 |
| POP (CVPR'23)[25] | 0.60 | 1.98 | 55.03 | **44.16** | 2.26 | 1.63 | 57.27 | **48.07** | 5.43 | 1.45 | 58.51 | **47.56** |
| **CLIP-DIN (ours)** | 3.92 | 21.37 | 59.82 | 53.48 | 6.65 | **14.21** | 60.63 | 57.89 | 9.83 | 11.10 | **59.63** | 76.31 |
| **DINOv2-DIN (ours)** | **4.04** | **21.92** | 59.83 | 49.68 | 5.67 | 13.58 | 58.10 | 53.05 | 10.86 | 11.12 | 58.94 | 80.57 |

Table 3: The performance comparison under the few-shot incremental setting.

| Method | GLide | DDPM | D-latent | DDIM | Pluralistic | repaint-p2 | Mean |
|---|---|---|---|---|---|---|---|
| LVNet[35] | 15.87 | 7.01 | 0.00 | 0.00 | 0.00 | 8.52 | 5.23 |
| POP[25] | 16.06 | 1.05 | 10.47 | 6.88 | 0.00 | 1.19 | 5.94 |
| **CLIP-DIN** | 15.39 | **9.39** | 10.25 | **18.89** | 0.31 | **18.42** | 12.11 |
| **DINO-DIN** | **26.68** | 6.31 | **12.42** | 13.18 | **2.65** | 16.12 | **12.89** |

Table 4: Results of the interpretation (by IOU) for known diffusion manipulated forgeries after Session 3 under the Few-shot incremental setting.

False Positive Rate (FPR). These superior results underline DIN's enhanced capability in interpreting both known and novel forgeries, benefitting significantly from its analysis of inter-forgery relationships and leveraging LVM-based semantic interventions.

We further explore DIN's adaptability across different Large Visual Models (LVMs), utilizing semantic features derived from models such as CLIP[32] and DINOv2[29], as detailed in the last two rows of Table 2. DIN consistently outperforms other methods when applied to various LVMs, demonstrating the effectiveness of our approach in leveraging purified semantic features across diverse self-training strategies and model architectures. This adaptability highlights DIN's robust capability to harness advanced semantic insights, enhancing its performance in open-world deepfake detection and interpretation tasks.

**Results under the few-shot incremental setting.** To ensure a comprehensive assessment suitable for real-world scenarios, we evaluated the proposed DIN against several state-of-the-art methods in the context of few-shot incremental setting. We utilize both $IOU_{novel}$ and $IOU_{old}$ to gauge incremental learning capabilities, and employ $IOU_{real}$ and FPR to assess the ability to identify novel forgeries. The results are revealed in Table 3, where traditional methods (fine-tuned LVNet[35] as well as POP[25] in first two rows) generally struggle to maintain consistent performance when learning new forgery technologies and detecting novel forgeries. Among these, POP [25] achieves the best average FPR in unknown forgeries but falls short in accurately localizing and attributing known forgeries. In contrast, our DIN demonstrates superior performance across most metrics, showcasing exceptional proficiency in recognizing newly added forgeries and generalizing the detection of unknown forgeries by leveraging inherent correlations and eliminating non-causal semantics.

To further evaluate the superiority of our proposed DIN in interpreting prevalent diffusion-based AIGCs, we list the IOU results for attribution and localization of seven diffusion techniques in Table 4. Our DIN surpasses competing methods in most categories, achieving the highest mean IOU with an average improvement of 6.95% by integrating DINOv2. These exceptional performances underscore

| LVM | SIM | SPM | Unknown | | | Known |
|---|---|---|---|---|---|---|
| | | | AUROC$\uparrow$ | AUPR$\uparrow$ | FPR$\downarrow$ | mACC$\uparrow$ |
| Baseline-DIN | $\times$ | $\times$ | 84.46 | 71.24 | 42.58 | 87.09 |
| DINO | $\checkmark$ | $\times$ | 86.04 | 74.31 | 42.71 | 87.95 |
| | $\checkmark$ | $\checkmark$ | **87.96** | **75.76** | **35.53** | 89.42 |
| CLIP | $\checkmark$ | $\times$ | 83.81 | 71.01 | 47.79 | **89.69** |
| | $\checkmark$ | $\checkmark$ | 85.86 | 72.10 | 37.39 | 89.37 |

Table 5: Ablation study on the effect of SPM and SIM with various LVMs.

DIN's effectiveness in leveraging a correlation-based incremental module and a semantic-prior orientation module, mitigating semantic overfitting in OW-DFI.

## 5.3 Ablation Study

**The effectiveness of SPM and SIM for interpretation.** We explore the impact of SPM and SIM on open-world interpretation and present the results in Table 5. Initially, DIN operates without SPM and SIM, as shown in the first row, where autonomously learned features struggle to differentiate between known and unknown forgeries due to the entanglement of forgery and non-causal semantics. Subsequently, incorporating comprehensive features derived from DINOv2 for semantic intervention (DINO+SIM) results in notable improvements of 1.58%, 3.07%, and 0.86% in AUROC, AUPR, and mACC, respectively. These results underscore the effectiveness of using LVM as a prior intervention to enhance the detection of unknown forgeries and improve the discrimination of known forgery techniques. Further integration of SPM (DINO+SIM+SPM) to refine and purify the semantics from the raw LVM features leads to additional improvements of 1.92%, 1.45%, and 1.47% in these metrics, alongside a significant 7.18% reduction in FPR. This underscores the inherent ambiguity in the LVM feature space and demonstrates the efficacy of SPM in refining semantics.

Additionally, we conduct an ablation study on the parameter $\lambda_{sil}$ during optimization to explore the effect of SIM. We observe that by increasing $\lambda_{sil}$ from 0.5 to 1.5 with a step of 0.5, DIN obtains the best experimental results under the 1.0 setting (85.86 of AUROC and 72.10 AUPR) compared to the 0.5 (84.86 AUROC and 70.94 AUPR) and 1.5 (85.52 AUROC and 71.26 AUPR) settings. This shows that SIM effectively improves the discriminative ability to detect unknown counterfeits. However, a more discriminant feature space caused by larger $\lambda_{sil}$ may be less in the relationships between forgeries, resulting in a slightly reduced effect of uncovering unknowns.

**The robustness to different LVMs.** To assess the robustness of our model to different LVMs, we conduct experiments using DINOv2 and CLIP, as shown in Table 5. Both DINOv2 and CLIP

| CIM | SOM | Known | | Unknown | |
|:---:|:---:|:---:|:---:|:---:|:---:|
| | | IOU$_{novel}$ | IOU$_{old}$ | IOU$_{real}$ | FPR ↓ |
| Session1 | | | | | |
| ✗ | ✗ | 0.00 | 21.42 | **61.39** | 62.66 |
| ✓ | ✗ | **4.10** | **21.42** | 59.83 | 62.77 |
| ✓ | ✓ | 3.92 | 21.37 | 59.82 | **53.48** |
| Session2 | | | | | |
| ✗ | ✗ | 0.00 | 8.75 | **61.80** | 58.46 |
| ✓ | ✗ | 6.32 | 13.34 | 60.31 | 59.68 |
| ✓ | ✓ | **6.65** | **14.21** | 60.63 | **57.89** |
| Session3 | | | | | |
| ✗ | ✗ | 0.00 | 2.02 | **61.57** | 77.12 |
| ✓ | ✗ | 8.89 | 10.58 | 59.64 | 77.83 |
| ✓ | ✓ | **9.83** | **11.10** | 59.63 | **76.31** |

**Table 6: Ablation study on the effect of CIM and SOM under few-shot incremental setting.**

demonstrate advanced semantic perception capabilities through their respective self-supervised learning paradigms. Without the Semantic Purification Module (SPM), relying on raw CLIP features significantly enhances the interpretation of known forgeries but slightly impairs the distinction of unknown forgeries. We attribute this to CLIP's limited fine-grained alignment capability, as it is primarily designed to correlate entire images with text descriptions [49]. In contrast, DINOv2 is trained to identify both image-level and patch-level resemblances, thus enhancing its utility for semantic interventions in Open-World Deepfake Interpretation (OW-DFI) with more small manipulated objects. Finally, the integration of SPM to refine semantic features significantly improves the detection of novel forgeries in CLIP, evidenced by a significant 10.4% reduction in FPR. These enhancements verify that our proposed DIN can effectively purify semantics from different LVM features, irrespective of their distinct training paradigms.

***The effectiveness of CIM and SOM.*** We investigate the impact of the proposed CIM and SOM under the few-shot incremental setting, as detailed in Table 6. Initially, the model uses basic loss functions, including localization loss $\mathcal{L}_{loc}$ and attribution loss $\mathcal{L}_{cls}$ as outlined in Section 4.1, to integrate new forgery knowledge. However, the baseline model struggles to capture nuanced features from limited samples, leading to suboptimal performance. The integration of CIM significantly enhances the transfer of existing forgery knowledge to new techniques, as evidenced by a 6.43% increase in IOU$_{novel}$ across three sessions. The addition of SOM further refines our approach by disentangling non-causal semantics, resulting in the most favorable outcomes: a 4.22% decrease in False Positive Rate (FPR) and a 0.37% improvement in IOU$_{novel}$. Notably, with the introduction of SOM, there is a significant 8.99% reduction in FPR during the first session, although IOU experienced a slight decline. However, across the incremental sessions, SOM achieves overall enhancements in all metrics. These results demonstrate that SOM can effectively minimize the model's overfitting to non-causal semantics, which is especially critical in densely populated category spaces, thus improving the robustness and accuracy of the model in recognizing and adapting to new forgeries.

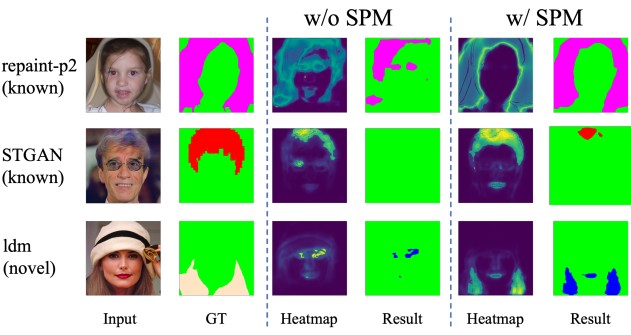

**Figure 3: The visualization of interpretation results, including forgery localization and attribution, based on the LVM feature (w/o SPM) and refined LVM semantics (w/ SPM), respectively. Deep blue signifies novel forgeries, green denotes the real regions, and other colors depict various recognized forgery categories.**

## 5.4 Visualization

To assess the efficacy of our proposed Semantic Purification module (SPM), we visualized the interpretation results by comparing the original and purified features from DINOv2, as depicted in Figure 3. In the w/o SPM columns, the identified forgery regions appear less precise and more dispersed, frequently extending into non-manipulated areas, which indicates a higher incidence of false positives. Conversely, the SPM-enhanced columns show significant improvements in both localization and attribution, particularly in detecting new types of forgeries. These findings underscore the effectiveness of SPM in enhancing OW-DFI by emphasizing the importance of LVM semantic purification and the removal of irrelevant semantics.

## 6 CONCLUSION

In this paper, we propose a novel task, Open-World Deepfake Interpretation (OW-DFI), designed to enhance the interpretability of deepfakes in dynamic open-world scenarios. It encompasses a wider variety of manipulated objects beyond just faces, addressing the complexities introduced by advanced forgery techniques. Our Deepfake Interpretation Network (DIN) leverages semantic insights from LVMs to mitigate reliance on spurious semantic patterns in a non-parametric manner. Moreover, a semantic-prior intervention strategy is developed to proactively exclude semantic priors from the deepfake identification process. In the incremental learning phase, we explore the relationships among different forgery technologies and enhance the distinction between semantic and forgery embeddings, which facilitates the minimization of semantic overfitting with limited training samples. Overall, our approach offers an innovative solution for deepfake detection by harnessing the fruitful semantic potential of LVMs and adapting to the dynamic advancements in forgery techniques within open-world contexts. Experimental results validate the SOTA performance of our method across 24 forgery methods, underscoring its capability to generate reliable interpretations and uncover novel forgeries as forgery techniques evolve.

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
