# OpenReview forum: "Boosting Non-causal Semantic Elimination: An Unconventional Harnessing of LVM for Open-World Deepfake Interpretation"
_acmmm.org/ACMMM/2024/Conference — MM2024 Poster_

### Official Review · Reviewer_AYwC · 2024-05-09

**Rating:** 4
**Confidence:** 2

**Summary:**

This paper proposes a new method aimed at enhancing the interpretation of deepfakes in open-world scenarios, called OW-DFI. This method leverages the semantic insights of LVMs to purify semantic information through an SPM to identify and locate deepfakes. The paper highlights the importance of discovering new types of forgeries in dynamic open-world environments and experimentally demonstrates the superior performance of this method in deepfake interpretation.

**Strengths:**

I agree that the paper addresses an interesting research problem. Stronger performance compared to baseline models.

**Limitations:**

Several key weaknesses are brought up in the reviews. This includes the following:

(1) Does the performance of the method come from large models such as CLIP or the design of the DIN architecture? Because in Table 5, Baselines-DIN does not specify whether it is DINO, CLIP, or what? If DIN is used as a framework and LVNet, POP, and OW-DFA are placed in this framework, will the performance be improved?

(2) Why is there no comparison with the OW-DFA baseline in Table 3 and Table 4?

(3) The experimental results lack standard deviation. Although most of the performances are significantly higher than existing methods, a small number of performances are quite different from existing methods. Is this caused by the randomness of deep learning or is it the superiority of the method? Therefore, multiple experiments with standard deviation or significance testing can make people more convinced of the superiority of certain indicators.

If you can provide reasonable answers to the above questions, I'd be happy to improve your score.

**Suitability:**

2

---

### Official Review · Reviewer_T5ni · 2024-05-20

**Rating:** 1
**Confidence:** 4

**Summary:**

This paper proposes a method to localize imperceptible artifacts across diverse manipulated objects and decipher forgery methods, so that handle a greater variety of manipulable objects and increasingly realistic artifacts. Specifically, the proposed method includes Semantic Intervention Learning (SIL) and Correlation-based Incremental Learning (CIL). To address the obstacle of complex visual effects in realistic and arbitrarily faked objects, SIL aims to disentangle and eliminate non-causal semantics by leveraging semantics derived from LVMs, while CIL combats catastrophic forgetting and semantic overfitting through an inter-forgery inheritance transpose and a targeted semantic intervention. Experimental results show the performance of the proposed method across different forgery methods.

**Strengths:**

1、The proposed method with the four modules looks quite reliable.
2、The idea is dedicated to identifying known forgery methods and localizing manipulated pixels.
3、To discover novel forgeries, the authors design a correlation-based incremental module in its incremental learning phase, facilitating knowledge transfer across various forgeries.

**Limitations:**

1、Figure II is confusing and difficult to understand, e.g., what is the Correlation-based FIL? Could the authors mark the modules mentioned later in the figure?
2、The general architecture of the method is not clear and contains many errors. The issues should not be limited to those listed below: First, in ‘4.1 The Architecture’, the meaning of ‘D^t’ (right column, line 443) is not given. Second, the use of symbol ‘A ̂(X_n )’ in Equation 1 and Equation 2 is inconsistent. Third, the authors do not explain the motivation for “produce a disrupted image” and the “disrupted image” does not appear in subsequent steps. Fourth, there is no explanation of how the “cost function” can be utilized for incremental learning.
3、 In the benchmark evaluation, the CLIP model is chosen to generate semantic features, but it is a visual language model. How did the authors achieve the generation? What are the implementation details of the comparison methods under the few-shot incremental setting?

**Suitability:**

3

---

### Official Review · Reviewer_82bU · 2024-05-21

**Rating:** 5
**Confidence:** 3

**Summary:**

This paper propose a new task for deepfake detection, i.e., the Open-World Deepfake Interpretation (OW-DFI). This task introduce the incremental learning into distinguish new deepfake methods, which is similar to the real world implementation scenario. Meanwhile, authors proposes a novel method for incremental OW-DFI with LVMs, which achieves great detection performance.

**Strengths:**

1. The proposed new task OW-DFI is close to real world implementation scenario
2. The proposed method will adapt the pretrained LVMs to OW-DFI with SIL and CIL
3. The paper writing is good and the introduction is clear. Moreover, the performance of proposed method achieves remarkable performance

**Limitations:**

1. Authors mention that the OW-DFI benchmark contains various content, including facial image, LSUN dataset and even animals. Moreover, it also includes various manipulation methods like CNN, GAN and Diffusion. I am curious about the cross-content and cross-manipulation detection performance, i.e. for the unseen content and manipulation methods.
2. For LVM's Semantic Purifcation, authors adapt a frequency domain augmentation. However, this kind of augmented images are unlike nature images that the LVM usually processes. Authors should explain why the frozen LVMs can well handle this augmented images.
3. As a face and deepfake related research, it is better to discuss the ethics issue in conclusion.

**Suitability:**

2

---

### Official Review · Reviewer_pcid · 2024-05-25

**Rating:** 5
**Confidence:** 3

**Summary:**

The paper introduces a novel task called open-world deepfake interpretation (OW-DFI), focusing on enhancing the interpretability of deepfake detection in scenarios where new deepfake methods from both familiar and new categories (e.g., diffusion-based ones) frequently emerge. To address this challenge, the authors propose a deepfake interpretation network (DIN). This network leverages non-causal semantics from large visual models (LVMs), such as CLIP and DINOv2. Specifically, the paper introduces two innovative techniques: semantic intervention learning, which enhances the detection of forgery artifacts by amplifying their inconsistencies with the "purified" semantics from LVMs, and correlation-based incremental learning, which mitigates catastrophic forgetting and semantic overfitting. Experimental results demonstrate the effectiveness of the proposed method for the OW-DFI task.

**Strengths:**

1. Elevating the task from open-world deepfake detection to open-world deepfake interpretation is a novel improvement that enhances the interpretation of deepfake detection.

2. The proposed techniques, semantic intervention learning and correlation-based incremental learning, are innovative and address key challenges in the field.

**Limitations:**

1. The writing is quite abstract and compact, making it difficult to understand the technical details behind the proposed method. If the 8-page limit restricts detailed descriptions, supplementary materials could be used to provide additional information.

2. The term "animals" in line 127 is unclear and needs clarification.

3. SIM and CIM are not properly defined, leaving readers to guess their meanings.

4. The role of Session 4 in the proposed OW-DFI benchmark is unclear, as it is never used and contains only one manipulation method.

**Suitability:**

3

---

### Meta-Review · Area_Chair_PxUn · 2024-07-06

**Recommendation:** Accept (Poster)
**Confidence:** 4

**Metareview:**

The paper introduces a new task called open-world deepfake interpretation (OW-DFI). OW-DFI involves localization of artifacts across manipulated objects and identification of forgery methods, including new forgery techniques. To address this new task, the paper proposes to leverage non-causal semantics from large visual models (LVMs). The paper introduces two techniques: semantic intervention learning, and correlation-based incremental learning.

Before rebuttal, the ratings are WA/WA/R/BA. Some concerns are writing and experiments e.g. comparison with open-world deepfake attribution (OW-DFA) baselines. The rebuttal is effective to address concerns. The ratings improve to WA/WA/BA/WA, all reviewers agree acceptance of the paper.

I recommend acceptance of the submission, and suggest authors to include new results in the camera-ready version.